# *In silico* design and immunoinformatics analysis of a universal multi-epitope vaccine against monkeypox virus

**Samira Sanami**[1], **Shahin Nazarian**[2], **Sajjad Ahmad**[3], **Elham Raeisi**[4], **Muhammad Tahir ul Qamar**[5], **Shahram Tahmasebian**[6], **Hamidreza Pazoki-Toroudi**[7,8], **Maryam Fazeli**[9], **Mahdi Ghatreh Samani**[10]*

1 Medical Plants Research Center, Basic Health Sciences Institute, Shahrekord University of Medical Sciences, Shahrekord, Iran, 2 Ming Hsieh Department of Electrical and Computer Engineering, University of Southern California, Los Angeles, CA, United States of America, 3 Department of Health and Biological Sciences, Abasyn University, Peshawar, Pakistan, 4 Cellular and Molecular Research Center, Basic Health Sciences Institute, Shahrekord University of Medical Sciences, Shahrekord, Iran, 5 Department of Bioinformatics and Biotechnology, Government College University Faisalabad, Faisalabad, Pakistan, 6 Department of Medical Biotechnology, School of Advanced Technologies, Shahrekord University of Medical Sciences, Shahrekord, Iran, 7 Physiology Research Center, Faculty of Medicine, Iran University of Medical Sciences, Tehran, Iran, 8 Department of Physiology, Faculty of Medicine, Iran University of Medical Sciences, Tehran, Iran, 9 WHO Collaborating Center for Reference and Research on Rabies, Pasteur Institute of Iran, Tehran, Iran, 10 Clinical Biochemistry Research Center, Basic Health Sciences Institute, Shahrekord University of Medical Sciences, Shahrekord, Iran

* ghatrehsamanimahdi@yahoo.com

**Data Availability Statement:** All relevant data are within the manuscript and its Supporting Information files.

## Abstract

Monkeypox virus (MPXV) outbreaks have been reported in various countries worldwide; however, there is no specific vaccine against MPXV. In this study, therefore, we employed computational approaches to design a multi-epitope vaccine against MPXV. Initially, cytotoxic T lymphocyte (CTL), helper T lymphocyte (HTL), linear B lymphocytes (LBL) epitopes were predicted from the cell surface-binding protein and envelope protein A28 homolog, both of which play essential roles in MPXV pathogenesis. All of the predicted epitopes were evaluated using key parameters. A total of 7 CTL, 4 HTL, and 5 LBL epitopes were chosen and combined with appropriate linkers and adjuvant to construct a multi-epitope vaccine. The CTL and HTL epitopes of the vaccine construct cover 95.57% of the worldwide population. The designed vaccine construct was found to be highly antigenic, non-allergenic, soluble, and to have acceptable physicochemical properties. The 3D structure of the vaccine and its potential interaction with Toll-Like receptor-4 (TLR4) were predicted. Molecular dynamics (MD) simulation confirmed the vaccine's high stability in complex with TLR4. Finally, codon adaptation and *in silico* cloning confirmed the high expression rate of the vaccine constructs in strain K12 of *Escherichia coli* (*E. coli*). These findings are very encouraging; however, *in vitro* and animal studies are needed to ensure the potency and safety of this vaccine candidate.

**Funding:** This study was financially supported by the Research Deputy of Shahrekord University of Medical Sciences with grant number 6411.

**Competing interests:** The authors have declared that no competing interests exist.

## Introduction

After more than two years of significant global economic and health impacts from COVID-19, outbreaks of the monkeypox virus (MPXV) have been reported in many countries around the world, raising new concerns about a new global pandemic [1]. World Health Organization (WHO) has reported 87,301 confirmed cases and 130 confirmed deaths of MPX in 111 countries as of May 3, 2023 continents [2]. The virus was named MPXV after it was discovered in monkeys in a Danish laboratory in 1985 [3]. The first human case was identified in a nine-month-old child from the Democratic Republic of the Congo (DRC) in 1970 [4]. MPXV has since become endemic in the DRC and has spread to other African countries. The first MPXV outbreak outside of Africa has been reported in the United States, following the shipment of infected African mammals from Ghana [5].

The MPXV is divided into two clades: the Congo Basin (central African) clade and the West African clade. The Congo Basin clade is more virulent, with a fatality rate of 106% compared to the West African clade, which has a fatality rate of 3.6% [6]. MPXV is a zoonotic virus that can be transmitted from animal to human and from human to human [7]. Animal-to-human transmission can occur through biting or scratching, consumption of game meat, and direct or indirect contact with body fluids or feces [8]. Transmission from human to human may occur via large respiratory droplets. Because respiratory droplets can only travel a few feet, prolonged face-to-face contact is required to spread the virus. MPXV can be transmitted through contact with an infected person's body fluids, feces, and urine, as well as infected clothing or bed linen [1]. The MPXV virus may be released in wastewater during rinsing, showering, or using the toilet [9]. The current outbreak of this virus among many gay men or men who have sex with other men raises the possibility of sexual transmission of the virus [10].

The symptoms of MPXV infection are similar to but milder than those of smallpox [11] and include fever, chills, headache, muscle aches, and fatigue. The main difference between this disease's symptoms and those of smallpox is that MPX causes lymph node swelling (lymphadenopathy) [1]. The incubation period of MPX is usually 7–14 days, but it can take up to 21 days [1]. The infected person develops skin rashes that resemble blisters 1 to 3 days (sometimes more) after the onset of fever. The majority of these rashes begin on the face and spread to other areas of the body [12]. MPXV infection is diagnosed based on the patient's history, clinical symptoms, and laboratory tests such as PCR, ELISA, western blot, and immunohistochemistry. The World Health Organization (WHO) recommends using qRT-PCR to detect viral DNA during acute infection of the MPXV [1].

MPXV belongs to the Poxviridae family's Orthopoxvirus genus [13]. The morphology of the MPXV reveals that virions are ovoid or brick-shaped particles enclosed by a geometrically corrugated lipoprotein outer membrane, as with other orthopoxviruses. The size range of the MPXV is 200 by 250 nm [14]. The MPXV genome is a linear double-stranded DNA with a length of about 197 kbp, consisting of hairpin loops, some open reading frames, and tandem repeats, and both ends contain inverted terminal repeats (ITRs) with a length of 10 kbp [15].

Vaccines have historically been the most effective tool for preventing and even eradicating infectious diseases. Vaccines can also cause herd effects, which result in protection even among those who have not been vaccinated, which can be especially useful for low-income individuals who are unable to access health care [16]. So far, no vaccine has been developed specifically to protect against MPX infection. The FDA approved ACAM2000 for use against smallpox in 2007. Since this vaccine is made from a live, replication-competent Vaccinia virus, there is a risk of serious side effects in vaccinated individuals [17]. On the other hand, Jynneos, a non-replicating modified *Vaccinia Ankara* virus vaccine, was approved by the FDA in 2019 for the prevention of both MPX and smallpox. Jynneos does not cause the production of live

virus in vaccinated individuals, making it safer than ACAM2000, particularly for use in immu-nocompromised individuals; however, there is limited information on Jynneos's effectiveness in preventing MPXV infection in humans [18].

The emergence of new infectious diseases necessitates the development of novel vaccine design methods. Conventional vaccine development methods are time-consuming and costly, requiring the cultivation of pathogenic microorganisms and the identification of their immu-nogenic components [19]. Bioinformatics methods and techniques have expanded exponen-tially over the years, recently influencing immunological studies. Reverse vaccinology is a computational approach that uses bioinformatics methods to identify new antigens from genome sequence information without the need to isolate and culture the pathogen, signifi-cantly accelerating vaccine development [20]. The design of multi-epitope vaccines using reverse vaccinology is a new and rapidly growing field. These vaccines have gotten a lot of attention because of their high specificity, safety, low-cost production, and simultaneous induction of cellular and humoral immune responses [21, 22]; the major drawback of these vaccines is their low immunogenicity [23–25], which is why it is suggested that protein-based adjuvants be included in the structure of these vaccines to solve this problem [21].

This study aims to design a multi-epitope vaccine against the MPXV using a reverse vacci-nology approach. The cell surface-binding protein and envelope protein A28 homolog were used as target proteins for epitope prediction in this study. The cell surface-binding protein and envelope protein A28 homolog are attractive targets for vaccine development for the fol-lowing reasons: the cell surface-binding protein is responsible for important biological func-tions of the MPXV, including virion attachment to the host cell and entry of the virus into the host cell [26]; the envelope protein A28 homolog is also required for virus entry into the host cell and cell-cell fusion [27]. To improve the vaccine candidate's immunogenicity, the cholera toxin B subunit (CTxB) sequence, which has been proven to act as a potential viral adjuvant, was incorporated into the vaccine construct [28–30]. We hope that the findings of this study will help in the fight against MPXV by leading to the development of a vaccine.

## Materials and methods

The sequential steps used in this study to design and evaluate a multi-epitope vaccine against the MPXV are depicted in Fig 1.

### Retrieval of protein sequences and identification of conserved regions

For each cell surface-binding protein and envelope protein A28 homolog, a hundred sequences in FASTA format were obtained from the NCBI database (https://www.ncbi.nlm.nih.gov/). Following the removal of irrelevant sequences, partial sequences, and sequences with ambiguous characters, multiple sequence alignment (MSA) was done using Clustal Omega (https://www.ebi.ac.uk/Tools/msa/clustalo/) with default parameters [31].

### Prediction and screening of T-cell and linear B-cell epitopes

Cytotoxic T lymphocyte (CTL) epitopes play an important role in the elimination of virus-infected cells, the promotion of cellular immunity, and the reduction of virus levels in the body [32]. The ProPred-I server (https://webs.iiitd.edu.in/raghava/propred1/index.html) was used to predict CTL epitopes from the cell surface-binding protein and envelope protein A28 homolog of MPXV. ProPred-I is an online web tool that predicts MHC binding sites in an antigenic sequence for 47 MHC class-I alleles using a matrix-based approach [33]. The adap-tive immune response relies heavily on helper T-lymphocytes. They play roles in the activation of B-cells and the killing of infected target cells, respectively [34]. The ProPred server (https://

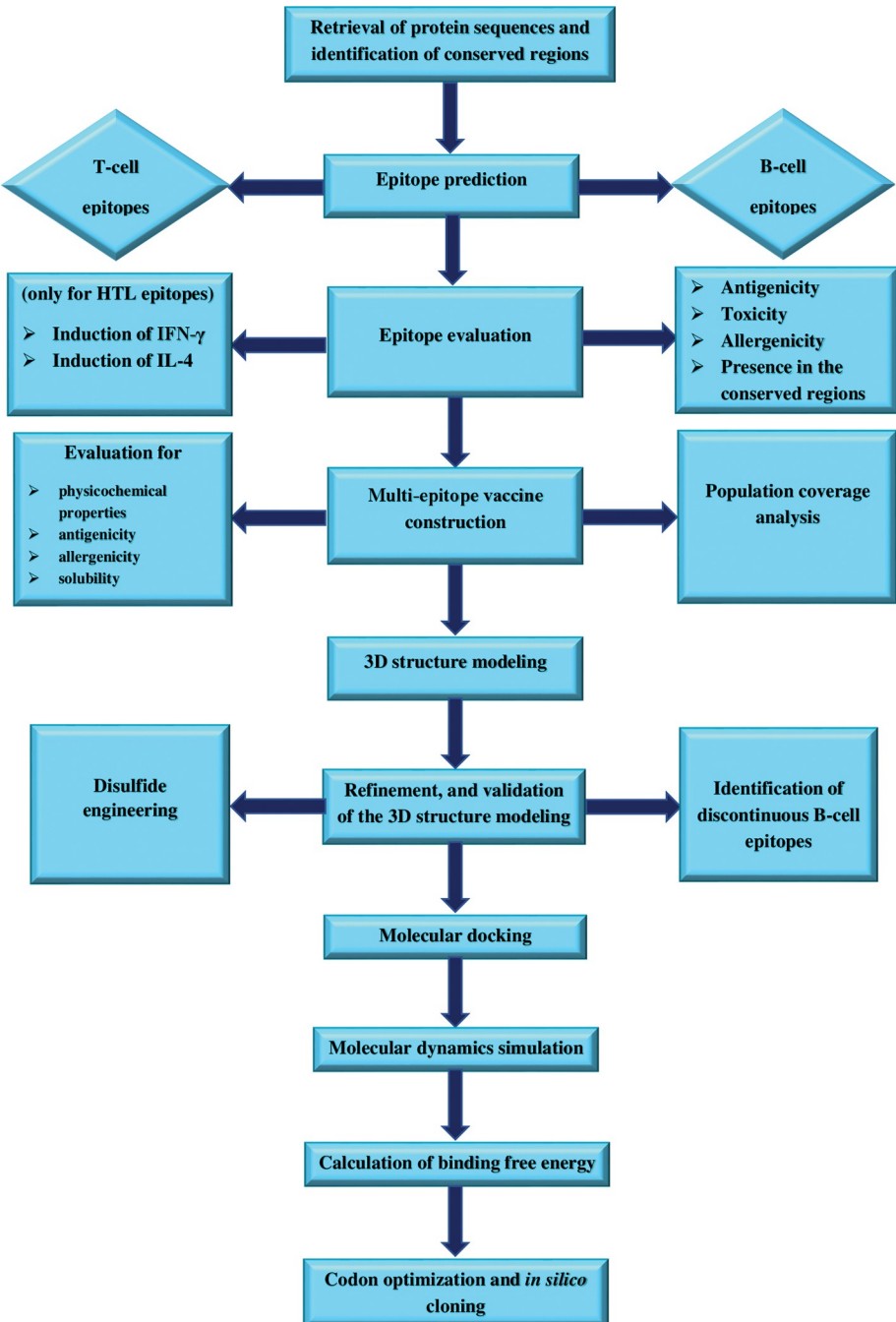

**Fig 1. Flowchart depicting the stages required in the *in silico* design of a multi-epitope vaccine against MPXV.** The process comprises numerous key phases, including target protein sequence retrieval, epitope prediction, vaccine construction, physicochemical characterization, 3D model prediction, molecular docking to immune receptor, MD simulation, and *in silico* cloning.

webs.iiitd.edu.in/raghava/propred/index.html) was used to predict the helper T lymphocyte (HTL) epitopes of the target proteins. ProPred predicts HTL epitopes in an antigen sequence using quantitative matrices for 51 HLA-DR alleles, which cover more than 90% of MHC Class II molecules expressed on antigen-presenting cells [35]. During a viral infection, B-cells use

viral epitopes to detect viruses, providing humoral immunity [36, 37]. The ABCpred server (https://webs.iiitd.edu.in/raghava/abcpred/ABC_submission.html) was used to identify linear B lymphocytes (LBL) epitopes. The ABCpred server predicts LBL epitopes with 65.93 percent accuracy using an artificial neural network (ANN) [38, 39].

The vaccine components, while antigenic and non-toxic, should not cause allergic reactions. As a result, the predicted epitopes were tested for antigenicity, toxicity, and allergenicity. VaxiJen 2.0 server (http://www.ddg-pharmfac.net/vaxijen/VaxiJen/VaxiJen.html) was used to calculate the antigenicity of the predicted epitopes. This is the first server for independent prediction of the level of protective antigens, allowing antigen classification based solely on the physicochemical properties of proteins without the use of sequence alignment [40–42]. In this study, we chose the virus as the target organism and set the antigenicity threshold at 0.4. The ToxinPred server (https://webs.iiitd.edu.in/raghava/toxinpred/design.php) was used to predict the potential toxicity of epitopes. A key feature of the server is that it calculates various physicochemical properties in addition to toxicity [43, 44]. The AllerTOP v. 2.0 server (https://www.ddg-pharmfac.net/AllerTOP/) was used to predict the allergenicity of the epitopes. To predict allergenicity, this server employs the auto cross covariance (ACC) transformation of amino acid sequences into uniform vectors of equal length [45]. The presence of predicted epitopes in the target proteins' conserved regions was then investigated. HTL epitopes that induce cytokines are important for vaccine development [46]. In addition to the previously mentioned parameters, HTL epitopes were tested for the production of cytokines such as interferon-gamma (IFN-γ) and interleukin-4 (IL-4). The two cytokines, IL-4 and IFN, play important roles in the generation and regulation of immune responses. IFN-γ promotes T helper type 1 responses, while IL-4 promotes T helper type 2 responses [47]. The IFNepitope server (https://webs.iiitd.edu.in/raghava/ifnepitope/design.php) was used to predict the ability of HTL epitopes to induce IFN-γ production [48]. In this study, we chose an SVM-based approach and an IFN-gamma versus other cytokine model for prediction. Furthermore, the IL4pred server (https://webs.iiitd.edu.in/raghava/il4pred/design.php) at 0.2 threshold was used to check HTL epitopes for IL-4 production [49].

## Multi-epitope vaccine construction

The final selected CTL, HTL, and LBL epitopes from the cell surface-binding protein and envelope protein A28 homolog were linked to each other using appropriate linkers to design a multi-epitope vaccine construct. The CTL, HTL, and LBL epitopes were connected using AAY, GGGGS, and KK linkers, respectively. Furthermore, to enhance the immunogenicity of the designed vaccine, the CTxB (accession number: P01556) was added as an adjuvant to the N-terminal of the vaccine construct via an EAAAK linker.

## Population coverage analysis

The frequency of various HLA alleles varies by geography and ethnic background [50, 51]. The IEDB population coverage analysis tool (http://tools.iedb.org/population/) was used to assess the population coverage of the vaccine candidate [52]. We used the selected CTL and HTL epitopes, as well as their MHC alleles, separately and in combination for this purpose. We additionally emphasized the total coverage of selected alleles across multiple continents.

## Prediction of antigenicity, allergenicity, solubility, and physicochemical properties of the vaccine construct

The antigenicity of the vaccine construct was predicted by the VaxiJen 2.0 server (threshold value of 0.4) and the ANTIGENpro server (http://scratch.proteomics.ics.uci.edu/).

ANTIGENpro predictions are sequence-based, alignment-free, and pathogen-independent. To make predictions, this server employs a two-stage architecture based on multiple representations of the primary sequence and five machine learning algorithms [53]. The AllerTOP v. 2.0 server was used to predict the allergenicity of the proposed vaccine. To test the solubility of the multi-epitope vaccine, the Protein-Sol server (https://protein-sol.manchester.ac.uk/) was used. The population average for the experimental dataset in this server is 0.45, so if the solubility value of a protein sequence is predicted to be greater than 0.45, it indicates that it has a higher solubility than the average soluble protein of *Escherichia coli* (*E. coli*) from the experimental solubility dataset [54]. Furthermore, the Expasy server's ProtParam tool (https://web.expasy.org/protparam/) was used to predict various physicochemical properties of the final vaccine construct, including the number of amino acids, molecular weight, theoretical pI, half-life, instability index, aliphatic index, and grand average of hydropathicity (GRAVY) [55].

### 3D structure modeling, refinement, and validation of the vaccine candidate

The I-TASSER server (https://zhanggroup.org/I-TASSER/) was used to predict the three-dimensional structure of the proposed vaccine. This server is a web-based resource for predicting protein structure and annotating structure-based functions. I-TASSER first recognizes structural templates from the PDB using multiple threading alignment approaches. Iterative fragment assembly simulations are then used to build full-length structural models. Finally, functional insights are obtained by matching predicted structure models with known proteins in function databases [56–58]. The predicted 3D model was refined to improve structural quality using the GalaxyRefine server (https://galaxy.seoklab.org/cgi-bin/submit.cgi?type=REFINEO). CASP10 has successfully tested the GalaxyRefine server's refining approach. The method first reconstructs side chains, then performs side chain repacking and overall structure relaxation using molecular dynamics simulation [59]. Validation is an important step in the modeling of proteins that includes evaluating the quality of the crude and refined models. We used the ZLab server (https://zlab.umassmed.edu/bu/rama/) and the ProSA-web server (https://prosa.services.came.sbg.ac.at/prosa.php) to assess the quality of the initial and refined models. The Ramachandran plot is generated by the Zlab server and reveals the two-dimensional distribution of the torsion angles—phi (φ) and psi (ψ)—of the amino acids contained in the protein backbone [60, 61]. The ProSA-web server computes an overall quality score for the protein structure. If this score falls outside of the normal range for native proteins, the structure most likely contains errors [62, 63].

### Identification of discontinuous B-cell epitopes

The ElliPro tool from the IEDB server (http://tools.iedb.org/ellipro/) was used to predict discontinuous B-cell epitopes in the vaccine's refined 3D model. ElliPro predicts and visualizes antibody epitopes in a given protein structure using Thornton's method, a residue clustering algorithm, the MODELLER program, and the Jmol viewer [64].

### Disulfide engineering of the multi-epitope vaccine

Disulfide engineering is a method for forming disulfide bonds in protein structures. Disulfide bonds promote the stability of the folded protein structure by decreasing conformational entropy and increasing the free energy of the denatured state [65]. The Disulfide by Design 2.13 server (http://cptweb.cpt.wayne.edu/DbD2/) was used to identify residue pairs in the vaccine construct that had the potential to mutate to cysteine and form a disulfide bond [66].

## Molecular docking of vaccine construct with TLR4

Molecular docking is an *in silico* method for predicting the binding affinity and orientation of a receptor and its ligand [67]. The Cluspro 2.0 server (https://cluspro.bu.edu/login.php) was used to perform molecular docking between the designed vaccine and Toll-Like receptor 4 (TLR4) (PDB ID: 4G8A). This server runs the following three computational steps to perform molecular docking: (1) rigid body docking is carried out using billions of conformations as samples, (2) clustering of the 1000 lowest energy structures using root-mean-square deviation (RMSD) to find the largest clusters that will represent the most likely models of the complex, and (3) chosen structures are refined using energy minimization [68–71]. The LigPlot software was used to generate schematic diagrams of bonds formed between vaccine construct residues and TLR4 [72].

## Molecular dynamics simulation of the TLR4-vaccine complex

Molecular dynamics (MD) simulation is a powerful computational method for determining the motions of proteins at the atomic scale over time, based on a general model of the physics governing interatomic interactions [73]. In this study, we used the GROMACS 2019.6 software to run an MD simulation [74]. The selected complex from the molecular docking step was used as input for the MD simulation process. The input structure was prepared using the ff99SB force field. This structure was solved in a cubic box of TIP3P water molecules using the gmx solvate software, and then $Na^+$ and $Cl^-$ ions were introduced to the protein's surface to neutralize the total charge of the system. The system's energy was minimized using the steepest descent algorithm. The system's temperature was gradually increased from 0 to 300 K for 200 ps at constant volume, and the system was then brought to equilibrium at constant pressure. The MD simulation was run at 300 K and for 40 ns.

## Calculation of binding free energy

The MMPBSA method and the gmx MMPBSA program were used to calculate the binding free energies for the vaccine, TLR4, and the vaccine-TLR4 complex. In this method, Poisson-Boltzmann (PB) and generalized Born (GB) models were used to calculate the binding free energy [75], and 1000 frames at regular intervals were extracted from the simulation trajectory and analyzed.

## Codon optimization and *in silico* cloning of the multi-epitope vaccine

The codon optimization approach can improve the expression efficiency of foreign genes in the host [76]. The Java Codon Adaptation Tool (JCat) (http://www.jcat.de/) was used to perform back translation, codon optimization, and determine the vaccine sequence's codon adaptation index (CAI) value and GC content [77]. In this analysis, the protein sequence of the designed vaccine was submitted to this server as input, and *E. coli* (strain K12) was chosen as the host organism [46]. Three additional options were selected to avoid the rho-independent transcription termination, prokaryote ribosome binding site, and cleavage sites of restriction enzymes. Following that, the *HindIII* and *BamHI* restriction enzyme recognition sequences were introduced to the N- and C-terminal of the vaccine sequence, respectively. The sequence was then inserted into the expression pET28a (+) vector using the SnapGene 3.2. 1software.

## Ethics statement

The ethical committee of Shahrekord University of Medical Sciences approved this study with the number: IR.SKUMS.REC.1401.065.

## Results

### Retrieval of protein sequences and identification of conserved regions

A total of seven appropriate sequences for the envelope protein A28 homolog and eighteen suitable sequences for the cell surface-binding protein were selected (S1 and S2 Data). The Clustal Omega then performed MSA to identify the conserved regions of target proteins. The conserved regions were selected based on the protein sequence's lack of gaps and the highest number of similar amino acids.

### Prediction and screening of T-cell and linear B-cell epitopes

Using the ProPred-I server, 78 CTL epitopes for the cell surface-binding protein and 56 CTL epitopes for the envelope protein A28 homolog were predicted. Among them, 12 CTL epitopes of the cell surface-binding protein and 21 CTL epitopes of the envelope protein A28 homolog that was capable of binding to at least three MHC class-I alleles were tested for antigenicity, toxicity, and allergenicity using the VaxiJen v2.0, ToxinPred, and AllerTOP v. 2.0 servers, respectively. The screened epitopes were then checked for their presence in the conserved regions of the target proteins. Finally, 2 CTL epitopes for the cell surface-binding protein (S1 Table) and 7 CTL epitopes for the envelope protein A28 homolog (S1 and S2 Tables) were chosen. Here, we predicted 38 HTL epitopes for the cell surface-binding protein and 25 HTL epitopes for the envelope protein A28 homolog using the ProPred server. Antigenicity, non-allergenicity, toxicity, presence in conserved regions, and IFN-γ and IL-4 production were assessed for 20 HTL epitopes of the cell surface-binding protein and 16 HTL epitopes of the envelope protein A28 homolog that could bind to at least three MHC class-II alleles. In total, 3 HTL epitopes were chosen for each target protein based on the mentioned parameters (S3 and S4 Tables). Moreover, the ABCpred server predicted 19 and 12 LBL epitopes for the cell surface-binding protein and envelope protein A28 homolog, respectively. After evaluation, 4 LBL epitopes for the cell surface-binding protein and 1 LBL epitope for the envelope protein A28 homolog with a score of 0.8 or higher were found to be antigenic, non-allergenic, and non-toxic in the conserved regions (S5 and S6 Tables).

### Multi-epitope vaccine construction

Overlapping epitopes were removed to avoid epitope duplication in the vaccine construct's linear structure. The final vaccine structure consisted of 1 adjuvant (CTxB), 7 CTL, 4 HTL, and 5 LBL epitopes linked together with EAAAK, AAY, GGGGS, and KK linkers, in that order (Fig 2). The epitopes that constituted the vaccine's structure are listed in Table 1.

### Population coverage analysis

To estimate population coverage of selected CTL and HTL epitopes, the IEDB population coverage analysis tool was employed. The CTL and HTL epitopes in the multi-epitope vaccine can cover 87.67% and 64.07% of the world population, respectively. The population coverage for combined CTL and HTL epitopes was estimated to be 95.57%. The population coverage of CTL and HTL epitopes when used in combination is highest in Europe (98.22%), followed by North America (96.66%), West Indies (95.48%), East Asia (94.37%), North Africa (94.24%), South America (94.13%), East Africa (93.5%), South Asia (92.55%), Central Africa (92.43%), West Africa (90.34%), Southwest Asia (90.15%), South Africa (84.98%), Northeast Asia (81.99%), Southeast Asia (80.06%), and Oceania (75.54%) (Fig 3).

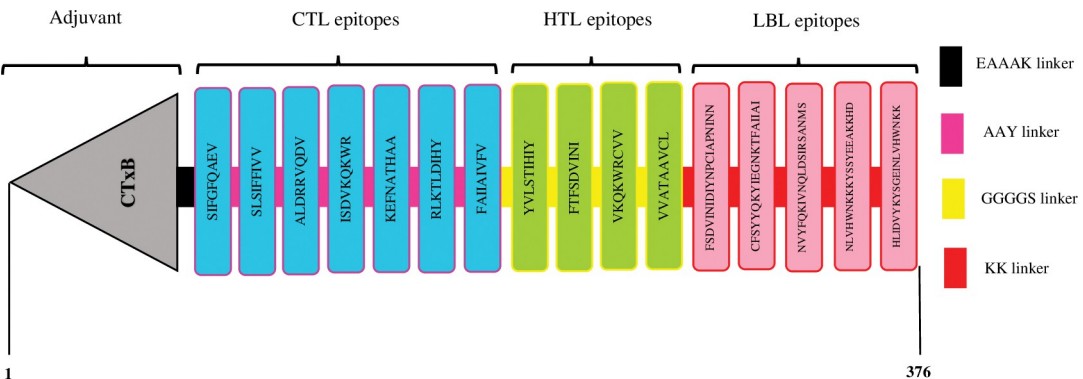

**Fig 2. A schematic representation of the multi-epitope vaccine construct against MPXV.** It includes 376 amino acids, 124 of which are adjuvant (grey). CTL, HTL, and LBL epitopes were represented by blue, green, and pink, respectively.

## Prediction of antigenicity, allergenicity, solubility, and physicochemical properties of the vaccine construct

The VaxiJen 2.0 server predicted the antigenicity of the designed vaccine to be 0.6355 with a virus model and 0.6107 with a bacteria model at a threshold of 0.4, while the ANTIGENpro server predicted the probability of antigenicity of the vaccine to be 0.774680. The vaccine construct was identified as non-allergen by the AllerTOP v.2.0 server. Furthermore, the Protein-Sol server predicted that our proposed vaccine was soluble, with a score of 0.455. The ProtParam tool predicted the physiochemical characteristics of the vaccine candidate, which are displayed in Table 2.

## 3D structure modeling, refinement, and validation of the vaccine candidate

I-TASSER server predicted the five 3D structure models of our vaccine construct using several threading templates (PDB Hit: 1LTR, 7P1A, 7LVB, 4L6T, 7QQK, 3KIG, 7Q0D, and 5ELB). The Z-score values ranged from 1 to 8.82, revealing that all of the threading templates were properly aligned; a Z-score greater than 1 indicates good alignment, and vice versa. Models 1–5 had C-scores of -1.29, -2.18, -2.45, -2.24, and -4.02, respectively. The C-score typically ranges from -5 to 2, with a higher C-score indicating a more confident model. As a result, we chose model 1 with a C-score of -1.29 for the refinement process (Fig 4). The GalaxyRefine server was used to refine the selected model, generating five refined models with different quality assessment parameters (Table 3). Higher GDT-HA and Rama favored values, as well as lower RMSD, MolProbity, Clash score, and Poor rotamers values, indicate that the models are of higher quality [78]. Model 2 was chosen for further evaluation based on the parameters

**Table 1. A list of the epitopes used in the proposed multi-epitope vaccine.**

| Protein | CTL epitopes | HTL epitopes | LBL epitopes |
|---|---|---|---|
| Cell surface-binding protein | RLKTLDIHY<br>FAIIAIVFV | YVLSTIHIY | CFSYYQKYIEGNKTFAIIAI<br>HLIDVYKYSGEINLVHWNKK<br>NVYFQKIVNQLDSIRSANMS<br>NLVHWNKKKYSSYEEAKKHD |
| Envelope protein A28 homolog | ALDRRVQDV<br>SLSIFFIVV<br>SIFGFQAEV<br>ISDVKQKWR<br>KEFNATHAA | FTFSDVINI<br>VKQKWRCVV<br>VVATAAVCL | FSDVINIDIYNPCIAPNINN |

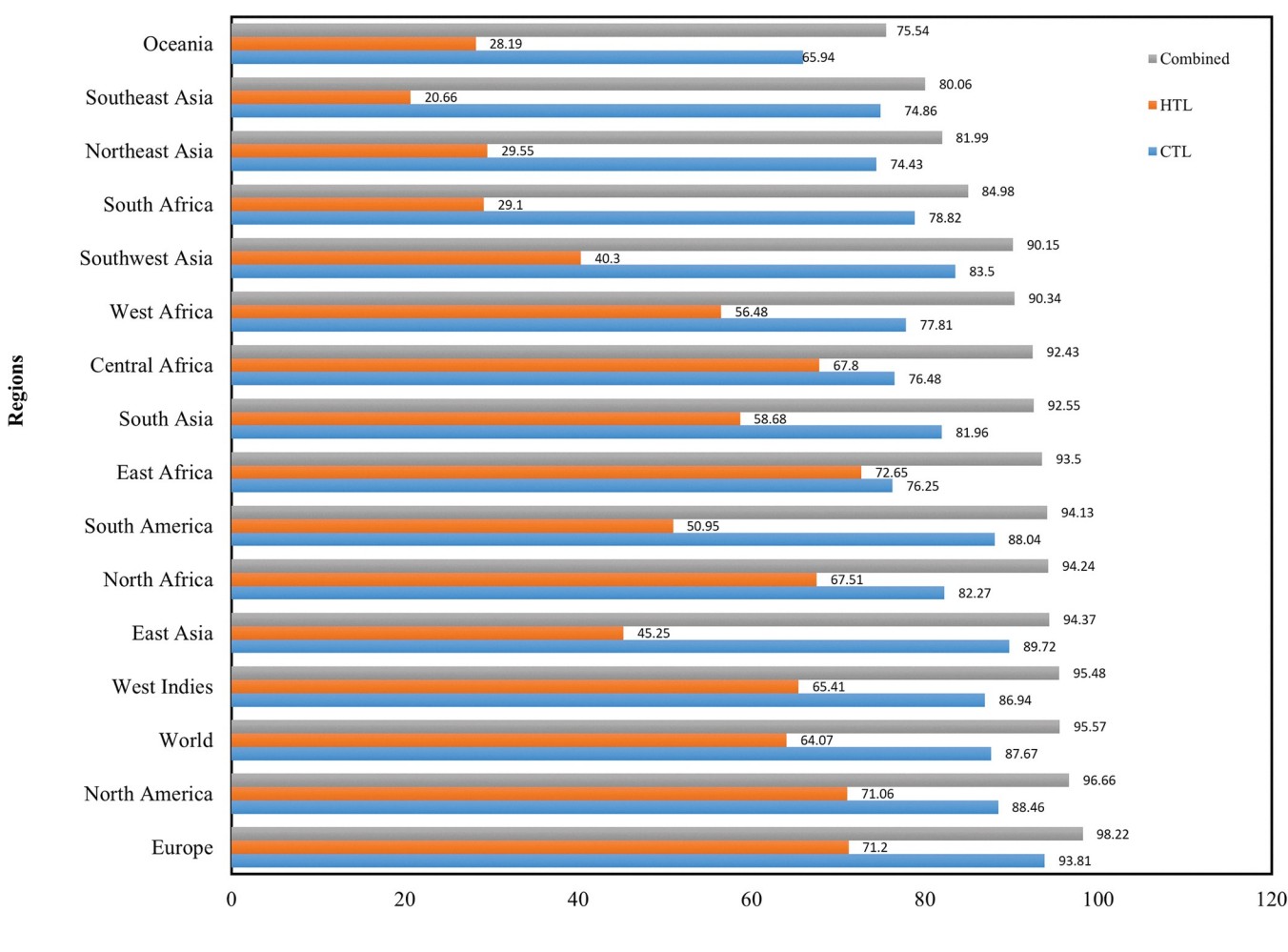

**Fig 3. Population coverage of selected CTL and HTL epitopes and their corresponding MHC alleles separately and in combination.**

mentioned above (Fig 4). The Zlab server and the ProSA-web server were used to analyze the overall quality of the vaccine construct's three-dimensional structure before and after refining. The Ramachandran plot analysis of the initial model revealed that 83.188%, 11.304%, and 5.507% of residues were present in the highly preferred, preferred, and questionable regions,

**Table 2. The physicochemical properties of a multi-epitope vaccine.**

| Parameter | Value |
|---|---|
| Number of amino acids | 376 |
| Molecular weight | 41.97 kDa |
| Theoretical pI | 9.48 |
| Half-life | 30 hours (mammalian reticulocytes, *in vitro*) >20 hours (yeast, *in vivo*) >10 hours (*E. coli*, *in vivo*) |
| Instability index | 38.70 |
| Aliphatic index | 91.12 |
| GRAVY | -0.044 |

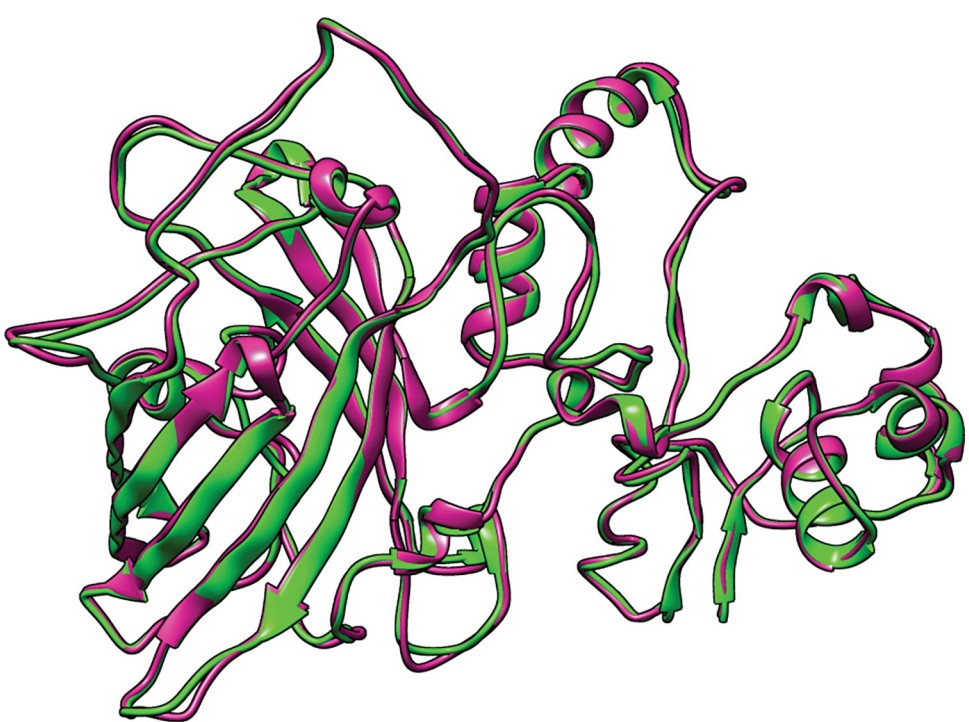

**Fig 4. The initial 3D structure of the multi-epitope vaccine is illustrated in green, and the refined 3D structure is presented in violet.** To compare the initial and refined models, the structures were superimposed.

respectively (Fig 5A), whereas, in the refined model, these values changed to 94.783%, 4.638%, and 0.580%, respectively (Fig 5B). The ProSA web server calculated the z-scores for the initial model to be -3.48 (Fig 5C), which changed to -3.67 after refinement (Fig 5D).

## Identification of discontinuous B-cell epitopes

The ElliPro server predicted eight discontinuous B-cell epitopes with scores greater than 0.5 in the vaccine construct (Fig 6). The predicted epitopes ranged in size from 5 to 79 amino acids (Table 4).

## Disulfide engineering of the multi-epitope vaccine

The Disulfide by Design 2.13 server identified 24 residue pairs in the vaccine construct's refined model that could form a disulfide bond (Table 5). The $\chi 3$ peaks in 1505 native disulfide bonds in 331 non-homologous proteins have been observed at -87 and +97 degrees and approximately 90% of naturally formed disulfide bonds have an energy value of less than 2.2

**Table 3. The quality parameter scores of the models generated by the GalaxyRefine server.**

| Model | GDT-HA | RMSD | MolProbity | Clash score | Poor rotamers | Rama favored |
|---|---|---|---|---|---|---|
| Initial | 1.0000 | 0.000 | 3.282 | 19.5 | 13.6 | 70.6 |
| MODEL 1 | 0.9162 | 0.501 | 2.379 | 19.5 | 0.0 | 88.2 |
| MODEL 2 | 0.9202 | 0.509 | 2.360 | 11.8 | 0.6 | 89.6 |
| MODEL 3 | 0.9122 | 0.514 | 2.438 | 19.5 | 1.3 | 89.0 |
| MODEL 4 | 0.9189 | 0.525 | 2.429 | 18.1 | 1.3 | 88.5 |
| MODEL 5 | 0.9169 | 0.518 | 2.431 | 19.5 | 1.3 | 89.3 |

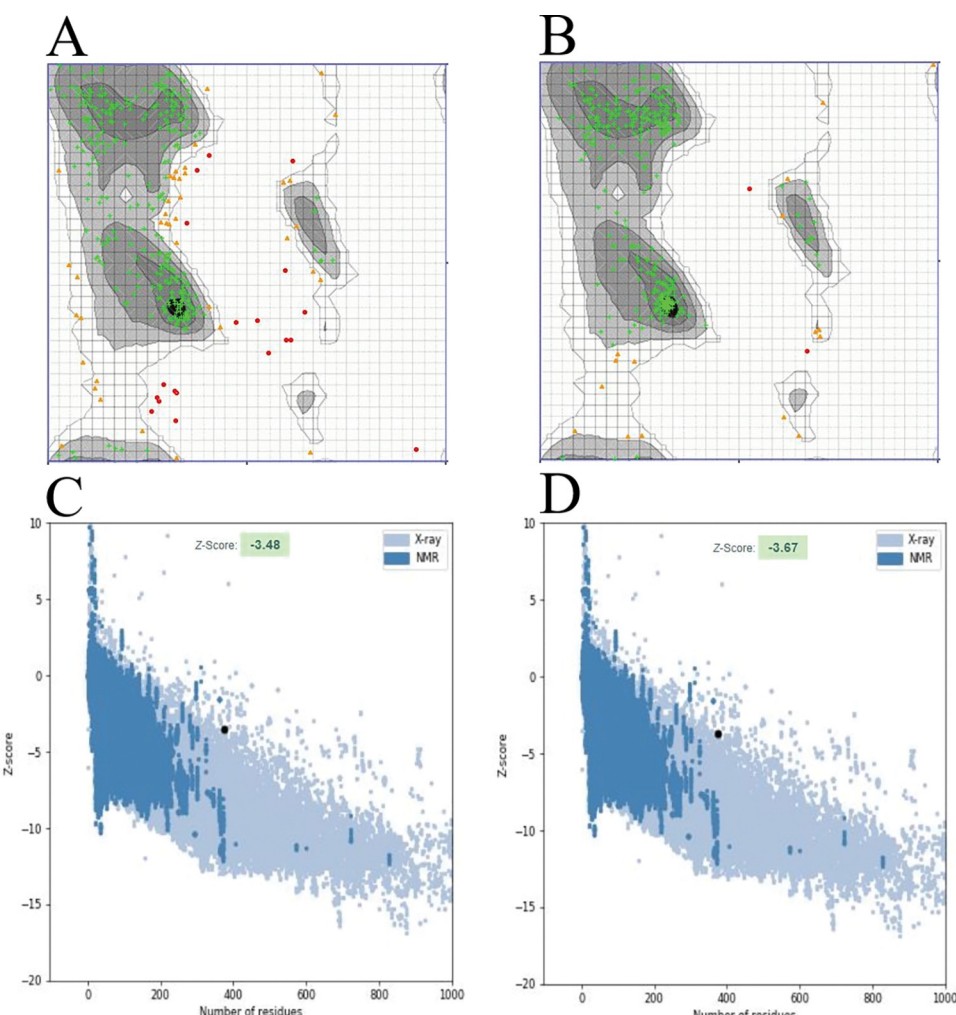

**Fig 5. Comparing the quality of the initial and refined models of the vaccine construct's three-dimensional structure using the Zlab server and the ProSA-web server.** (A) The Ramachandran plot analysis shows that in the initial model, 83.188%, 11.304%, and 5.507% of the residues are found in the highly preferred (green crosses), preferred (brown triangles), and questionable (red circles) regions, respectively; (B) whereas in the refined model, 94.783%, 4.638%, and 0.580% of the residues are found in the highly preferred, preferred, and questionable regions, respectively. (C) The initial model has a Z-score of -3.48, (D) while the refined model has a Z-score of -3.67.

kcal/mol [66]. Based on a χ3 angle between -87˚ and +97˚ and an energy value less than 2.2 kcal/mol, only three residue pairs were chosen for disulfide bond formation. These residue pairs were LYS 290-ALA 306, PHE 146-LYS 341, and SER 345-ALA 350 (Fig 7).

## Molecular docking of vaccine construct with TLR4

Molecular docking between the vaccine and TLR4 was done using the Cluspro 2.0 server. This server generated 11 clusters and calculated the number of members and the corresponding lowest energy for each cluster. The comparison of all clusters revealed that cluster 1 has the strongest interaction energy (the most negative) with the lowest energy of -999.8 kcal/mol and 29 members (Fig 8). The LigPlot tool illustrated the bonds formed between the amino acids in the vaccine and TLR4 (Fig 9). The amino acids involved in hydrogen bonds, as well as their lengths, are listed in Tables 6 and 7.

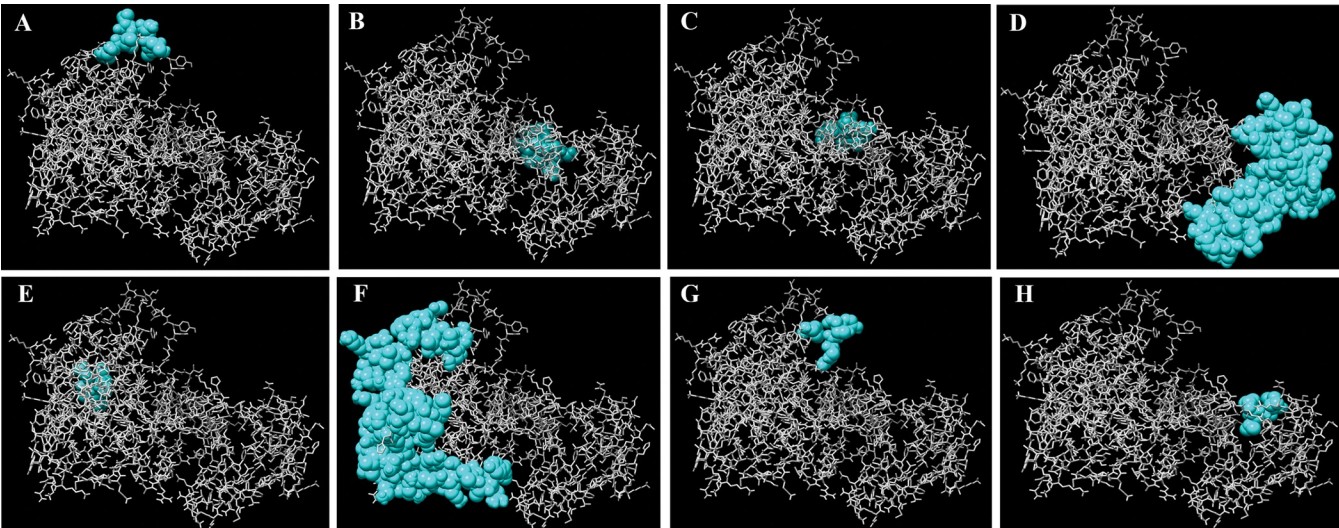

**Fig 6.** The discontinuous B-cell epitopes in the multi-epitope vaccine 3D model (A-H). The vaccine construct is depicted by gray sticks, while the discontinuous B-cell epitopes are shown by cyan spheres. (A) 8 residues with a score of 0.948; (B) 11 residues with a score of 0.833; (C) 6 residues with a score of 0.744; (D) 73 residues with a score of 0.732; (E) 16 residues with a score of 0.646; (F) 79 residues with a score of 0.646; (G) 6 residues with a score of 0.562; (H) 5 residues with a score of 0.515.

## Molecular dynamics simulation of the TLR4-vaccine complex

The MD simulation results obtained from GROMACS 2019.6 software were analyzed as RMSD and RMSF using gmx rms and gmx rmsf modules, respectively. The RMSD parameter of the structures formed during MD simulation in the time dimension is an appropriate and

**Table 4.** A list of the discontinuous B-cell epitopes predicted by the ElliPro server.

| No. | Residues | Number of residues | Score |
|---|---|---|---|
| 1 | A:V217, A:L218, A:S219, A:T220, A:I221, A:H222, A:I223, A:Y224 | 8 | 0.948 |
| 2 | A:V73, A:P74, A:G75, A:S76, A:Q77, A:H78, A:I79, A:D80, A:S81, A:Q82, A:K83 | 11 | 0.833 |
| 3 | A:K84, A:A85, A:I86, A:E87, A:R88, A:K90 | 6 | 0.744 |
| 4 | A:M1, A:I2, A:K3, A:L4, A:K5, A:F6, A:G7, A:V8, A:F9, A:V12, A:L13, A:L14, A:S15, A:S16, A:A17, A:Y18, A:A19, A:H20, A:G21, A:T22, A:P23, A:Q24, A:N25, A:I26, A:T27, A:D28, A:L29, A:C30, A:A31, A:E32, A:Y33, A:H34, A:N35, A:T36, A:Q37, A:I38, A:Y39, A:T40, A:L41, A:N42, A:D43, A:K44, A:S47, A:Y48, A:E50, A:S51, A:L52, A:A53, A:G54, A:K55, A:R56, A:E57, A:M58, A:A59, A:I60, A:I61, A:T62, A:K356, A:Y362, A:K363, A:Y364, A:S365, A:G366, A:E367, A:I368, A:N369, A:L370, A:V371, A:H372, A:W373, A:N374, A:K375, A:K376 | 73 | 0.732 |
| 5 | A:G133, A:F134, A:Q135, A:A136, A:E137, A:V169, A:K170, A:Q171, A:K172, A:R174, A:A175, A:A176, A:Y177, A:G301, A:K303, A:F305 | 16 | 0.646 |
| 6 | A:V149, A:V150, A:A151, A:A152, A:Y153, A:A154, A:R158, A:L191, A:K192, A:T193, A:L194, A:D195, A:I196, A:Y198, A:A199, A:A200, A:Y201, A:F202, A:G225, A:G226, A:G227, A:G228, A:S229, A:F230, A:T231, A:F232, A:S233, A:D234, A:V235, A:N237, A:G240, A:G241, A:G242, A:S243, A:V244, A:K245, A:K247, A:W248, A:R249, A:C250, A:V251, A:V252, A:G254, A:G255, A:V258, A:V259, A:A260, A:T261, A:A262, A:K267, A:F269, A:S270, A:D271, A:V272, A:I273, A:N274, A:I275, A:D276, A:I277, A:Y278, A:Y294, A:I308, A:A309, A:I310, A:K311, A:K312, A:N313, A:V314, A:Y315, A:F316, A:K318, A:S328, A:A329, A:N330, A:M331, A:S332, A:K333, A:K334, A:N335 | 79 | 0.646 |
| 7 | A:G212, A:G213, A:G214, A:S215, A:Y216, A:Y298 | 6 | 0.562 |
| 8 | A:H357, A:L358, A:I359, A:D360, A:V361 | 5 | 0.515 |

**Table 5. List of residue pairs capable of forming disulfide bonds in the vaccine construct.**

| Residue 1 | | | Residue 2 | | | Bond | | |
|---|---|---|---|---|---|---|---|---|
| Chain | Seq | AA | Chain | Seq | AA | χ3 | Energy (kcal/mol) | Σ B-factor |
| A | 4 | LEU | A | 39 | TYR | 74.49 | 6.7 | 0 |
| A | 4 | LEU | A | 62 | THR | -94.4 | 6.76 | 0 |
| A | 17 | ALA | A | 22 | THR | -88.6 | 2.53 | 0 |
| A | 63 | PHE | A | 68 | ILE | -108.72 | 4.43 | 0 |
| A | 78 | HIS | A | 82 | GLN | 78.35 | 4.41 | 0 |
| A | 110 | ASN | A | 348 | GLU | -79.67 | 3.31 | 0 |
| A | 114 | PRO | A | 119 | ALA | 75.29 | 5.94 | 0 |
| A | 120 | ILE | A | 211 | GLY | 125.8 | 2.4 | 0 |
| A | 123 | ALA | A | 350 | ALA | -72.61 | 4.31 | 0 |
| A | 124 | ASN | A | 345 | SER | -104.43 | 5.36 | 0 |
| A | 125 | GLU | A | 144 | SER | -81.96 | 4.36 | 0 |
| A | 146 | PHE | A | 341 | LYS | 101.7 | 1.74 | 0 |
| A | 152 | ALA | A | 158 | ARG | -79 | 2.72 | 0 |
| A | 179 | GLU | A | 302 | ASN | 84.18 | 6.41 | 0 |
| A | 183 | THR | A | 211 | GLY | -65.2 | 5.17 | 0 |
| A | 207 | ILE | A | 293 | SER | 111.2 | 4.16 | 0 |
| A | 230 | PHE | A | 259 | VAL | -79.15 | 4.45 | 0 |
| A | 240 | GLY | A | 321 | ASN | -63.48 | 4.1 | 0 |
| A | 257 | SER | A | 292 | PHE | 116.47 | 2.67 | 0 |
| A | 284 | PRO | A | 324 | ASP | -86.41 | 3.44 | 0 |
| A | 287 | ASN | A | 320 | VAL | 88.4 | 2.57 | 0 |
| A | 290 | LYS | A | 306 | ALA | 102.78 | 0.36 | 0 |
| A | 345 | SER | A | 350 | ALA | -72.56 | 2.02 | 0 |
| A | 351 | LYS | A | 354 | ASP | 126.83 | 5.21 | 0 |

widely used measure for assessing the structural stability of the protein [79]. The RMSD plot of the TLR4 indicated values in the 0.16–0.52 nm range. The vaccine's RMSD plot showed an upward trend at the beginning of the simulation, peaking at 0.84 nm after about 33 ns and fluctuating slightly until the simulation ended (Fig 10A). RMSF is a calculation of protein structure

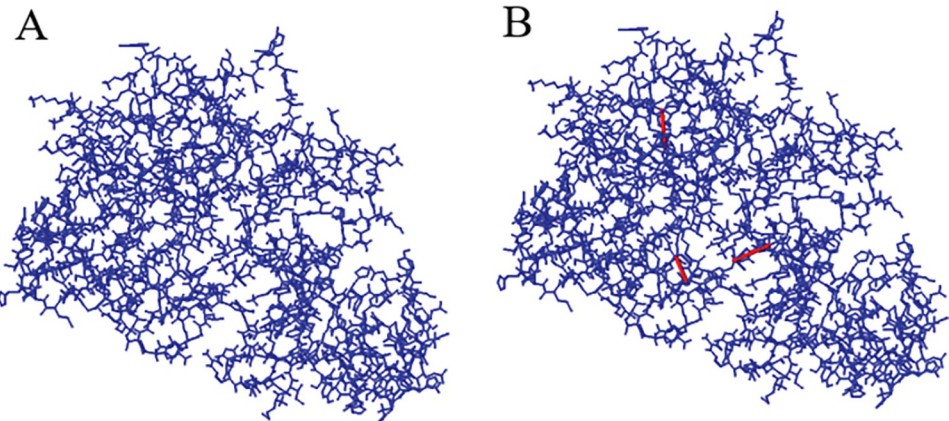

**Fig 7. The disulfide engineering of the multi-epitope vaccine's 3D structure.** (A) The wild type; (B) The mutant type (the three introduced disulfide bonds are shown by red sticks).

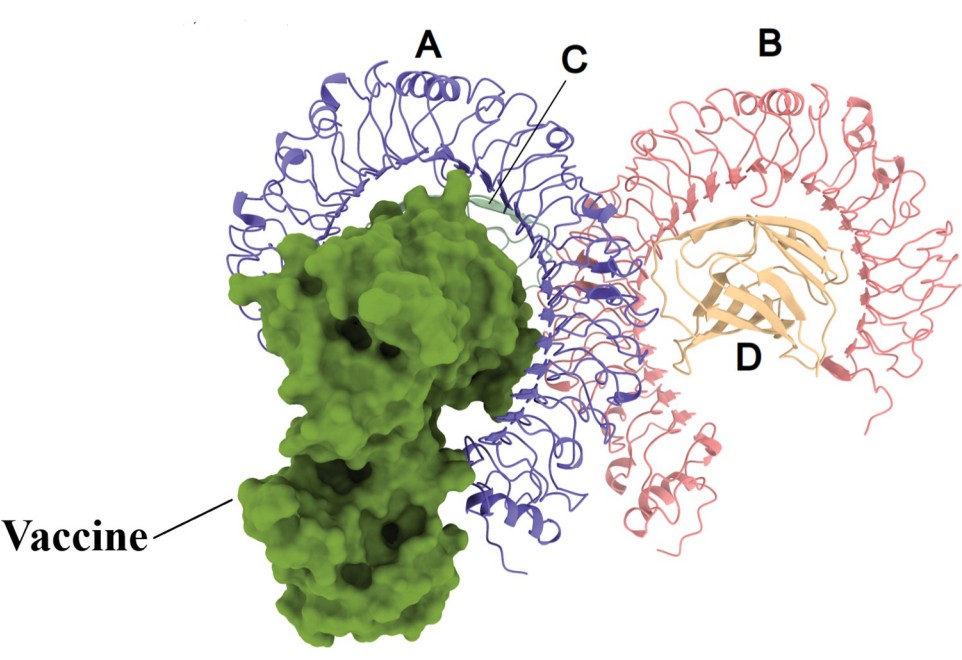

**Fig 8. The docked complex of the vaccine and TLR4.** The vaccine construct is shown in surface form, while TLR4 is shown in cartoon form.

residue fluctuation during an MD simulation. Because the two chains A and B from the TLR4 have the same sequence, the RMSF plot of these two chains is shown to more accurately study the effect of the vaccine construct binding on the flexibility of chain A. The residues at both ends of chain A have shown less flexibility than chain B, while the flexibility of other regions in the two chains is the same and does not show a noticeable change. The vaccine's RMSF plot revealed that the majority of the vaccine's residues were highly flexible (Fig 10B).

## Calculation of binding free energy

The binding free energies of TLR4, vaccine, and TLR4-vaccine complex were calculated using the MM-GBSA and MM-PBSA methods. In the MM-PBSA analysis, the total binding free energy was -53718.65 kcal/mol for TLR4, -33558.07 kcal/mol for the vaccine, and -87411.4 kcal/mol for the TLR4-vaccine complex. Moreover, the total binding free energy in the MM-GBSA was -54463.47 kcal/mol for TLR4, -34327.63 kcal/mol for the vaccine, and -88867.65 kcal/mol for the TLR4-vaccine complex. According to the calculated values, the gas phase energy contribution was equal and substantial in both methods. The contribution of each energy component is shown in Table 8.

## Codon optimization and *in silico* cloning of the multi-epitope vaccine

The vaccine protein sequence was reverse translated into a nucleotide sequence with a length of 1128 bp using the JCat server. This server estimated the optimized sequence's GC content to be 45.30% and its CIA value to be 0.97. The vaccine nucleotide sequence was then *in silico* cloned into the multiple cloning site (MCS) of pET28a (+) between *HindIII* (173) and *BamHI* (1307) restriction sites, resulting in a recombinant plasmid with a length of 6478 bp (Fig 11).

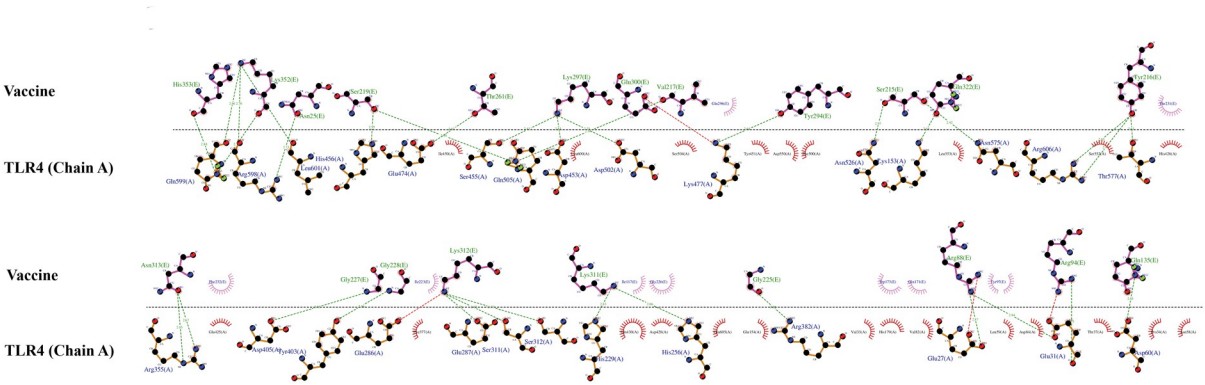

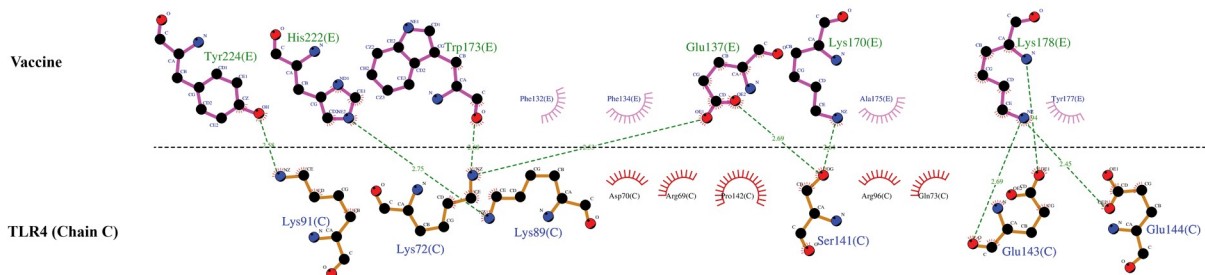

**Fig 9. The interaction map of the vaccine construct with chains A and C from TLR4.** Hydrogen bonds are shown by dashed (green) lines with the length of the bond printed in the middle.

## Discussion

Despite the fact that the MPXV is endemic in West and Central African countries, outbreaks in other countries with no known epidemiological links to West or Central Africa have occurred [80]. Because the MPXV is so closely related to smallpox, the smallpox vaccine is at least 85% effective in preventing MPX, but a vaccine that specifically targets the MPXV is needed [81]. The multi-epitope vaccines have a high potential to elicit humoral and cellular immune responses; these vaccines are composed of epitopes rather than large proteins or whole genomes, protecting the host from excessive antigenic load and allergic reactions [82]. In recent years, a large number of multi-epitope vaccines against various pathogens have been designed, including human papillomavirus (HPVs) [83], Zika virus [84, 85], Ebola virus [86, 87], SARS-CoV-2 [88–92], *Klebsiella pneumonia* [93], *Helicobacter pylori* [79, 94], *Staphylococcus aureus* [95], *Leishmania* [96, 97], and Cystic echinococcosis [98].

Several studies on multi-epitope vaccine design against MPXV have been carried out so far. In the study conducted by Aiman *et al*. (2022), selected epitopes from MPXVgp167, MPXVgp028, and MPXVgp105 proteins along with 4 different adjuvants (HBHA protein, β-defensins, 50S ribosomal protein L7/L12 adjuvants, and HBHA conserved peptide sequences) were included in four vaccine constructs [99]. In the study of Ullah *et al*. (2022), to design a multi-epitope vaccine against MPXV, three extracellular proteins, including a cupin domain-containing protein, ABC transporter ATP-binding protein, and DUF192 domain-containing

**Table 6. List of residues involved in the formation of hydrogen bonds between TLR4 (chains A) and the vaccine.**

| TLR4 (A) | Vaccine (E) | Bond Length (Å) | |
|---|---|---|---|
| Gln599 | His353 | 2.88 | |
| | Lys352 | 2.66 | 2.7 |
| Arg598 | Lys352 | 2.7 | |
| | Asn25 | 2.73 | |
| Leu601 | Lys352 | 2.45 | |
| His456 | Ser219 | 2.95 | |
| Glu474 | Thr261 | 2.87 | |
| Ser455 | Lys297 | 2.43 | |
| Gln505 | Ser219 | 2.94 | |
| | Val217 | 2.88 | |
| Asp453 | Lys297 | 2.85 | |
| Asp502 | Lys297 | 2.75 | |
| Lys477 | Tyr294 | 2.64 | |
| Asn526 | Ser215 | 2.97 | |
| Lys153 | Gln322 | 2.68 | |
| Asn575 | Ser215 | 2.92 | |
| Arg606 | Tyr216 | 2.65 | 2.97 |
| Thr577 | Tyr216 | 2.63 | |
| Arg355 | Asn313 | 2.85 | 2.67 |
| Asp405 | Gly227 | 2.95 | |
| Tyr403 | Gly228 | 2.96 | |
| Glu287 | Lys312 | 2.78 | |
| Ser311 | | 2.62 | |
| Ser312 | | 2.77 | |
| His229 | Lys311 | 2.83 | |
| His256 | | 2.66 | |
| Arg382 | Gly225 | 2.66 | |
| Glu27 | Arg88 | 2.87 | |
| Glu31 | | 2.84 | |
| Glu31 | Arg94 | 2.75 | |
| Asp60 | Gln135 | 2.85 | |

protein were selected as epitope prediction targets, as well as the CTxB was used as an adjuvant in the vaccine construct [100]. The multi-epitope vaccine designed by Shantier *et al.* (2022) included T-cell and B-cell epitopes selected from the cell surface-binding proteins, as well as

**Table 7. List of residues involved in the formation of hydrogen bonds between TLR4 (chain C) and the vaccine.**

| TLR4 (C) | Vaccine (E) | Bond Length (Å) | |
|---|---|---|---|
| Lys91 | Tyr224 | 2.58 | |
| Lys89 | His222 | 2.75 | |
| Lys72 | Trp173 | 2.68 | |
| | Glu137 | 2.53 | |
| Ser141 | Glu137 | 2.69 | |
| | Lys170 | 2.64 | |
| Glu143 | Lys178 | 2.94 | 2.69 |
| Glu144 | Lys178 | 2.45 | |

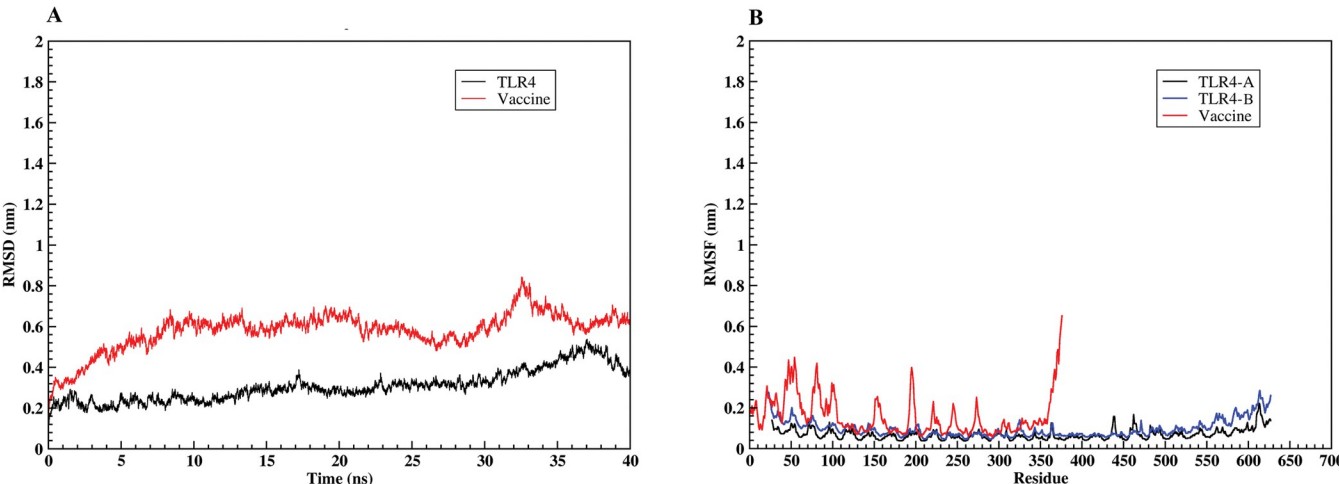

**Fig 10. The MD simulation of the vaccine-TLR4 docked complex in a period of 40 ns.** (A) The RMSD plot of TLR4 exhibits a consistent trend during the simulation period, while the vaccine's RMSD plot begins with an infinitesimal variation and becomes stable at about 33 ns. (B) The RMSF plot shows that both TLR4 chains A and B show very mild fluctuations, while the vaccine construct shows high fluctuations.

RS09 and PADRE adjuvants attached to the vaccine construct's N-terminal to increase immunogenicity [101]. Waqas *et al.* (2023) assembled the best B- and T-cell epitopes obtained from A35R, A43R, B16R, B21R, C4L, E8L, E13L, E21R, and G10R proteins into a vaccine construct and used RpfE sequence as an adjuvant to boost vaccine immunogenicity [102]. In the study by Zaib *et al.* (2023), two vaccine constructs were developed, one containing B- and T-cell epitopes from the COP-B7R protein and the other containing B- and T-cell epitopes from the COP-A44L protein. The 50 S ribosomal L7/L12 was used as an adjuvant at the N-terminal of both vaccine constructs [103]. Tan *et al.* (2023) designed five vaccine constructs based on B- and T-cell epitopes found in A35R, B6R, and H3L proteins. All other components of these constructs were identical except for the adjuvant, and five distinct adjuvants (50S ribosomal

**Table 8. Binding free energies of TLR4, vaccine, and TLR4-vaccine complex.** All values are given in kcal/mol.

| Energy component | TLR4 | Vaccine | TLR4-Vaccine Complex |
|---|---|---|---|
| MM-PBSA | | | |
| Van der Waals | -4965.78 | -2516.34 | -7689.16 |
| Electrostatic | -41692.74 | -25030.73 | -70148.83 |
| Polar solvation | -7190.2 | -6126.71 | -9790.73 |
| Non-polar solvation | 130.07 | 115.71 | 217.32 |
| Delta Gas phase | -46658.52 | -27547.07 | -77837.99 |
| Delta Solvation phase | -7060.13 | -6011 | -9573.41 |
| Delta Total | -53718.65 | -33558.07 | -87411.4 |
| MM-GBSA | | | |
| Van der Waals | -4965.78 | -2516.34 | -7689.16 |
| Electrostatic | -41692.74 | -25030.73 | -70148.83 |
| Polar solvation | -8000.85 | -6948.02 | -11364.64 |
| Non-polar solvation | 195.9 | 167.46 | 334.98 |
| Delta Gas phase | -46658.52 | -27547.07 | -77837.99 |
| Delta Solvation phase | -7804.95 | -6780.56 | -11029.66 |
| Delta Total | -54463.47 | -34327.63 | -88867.65 |

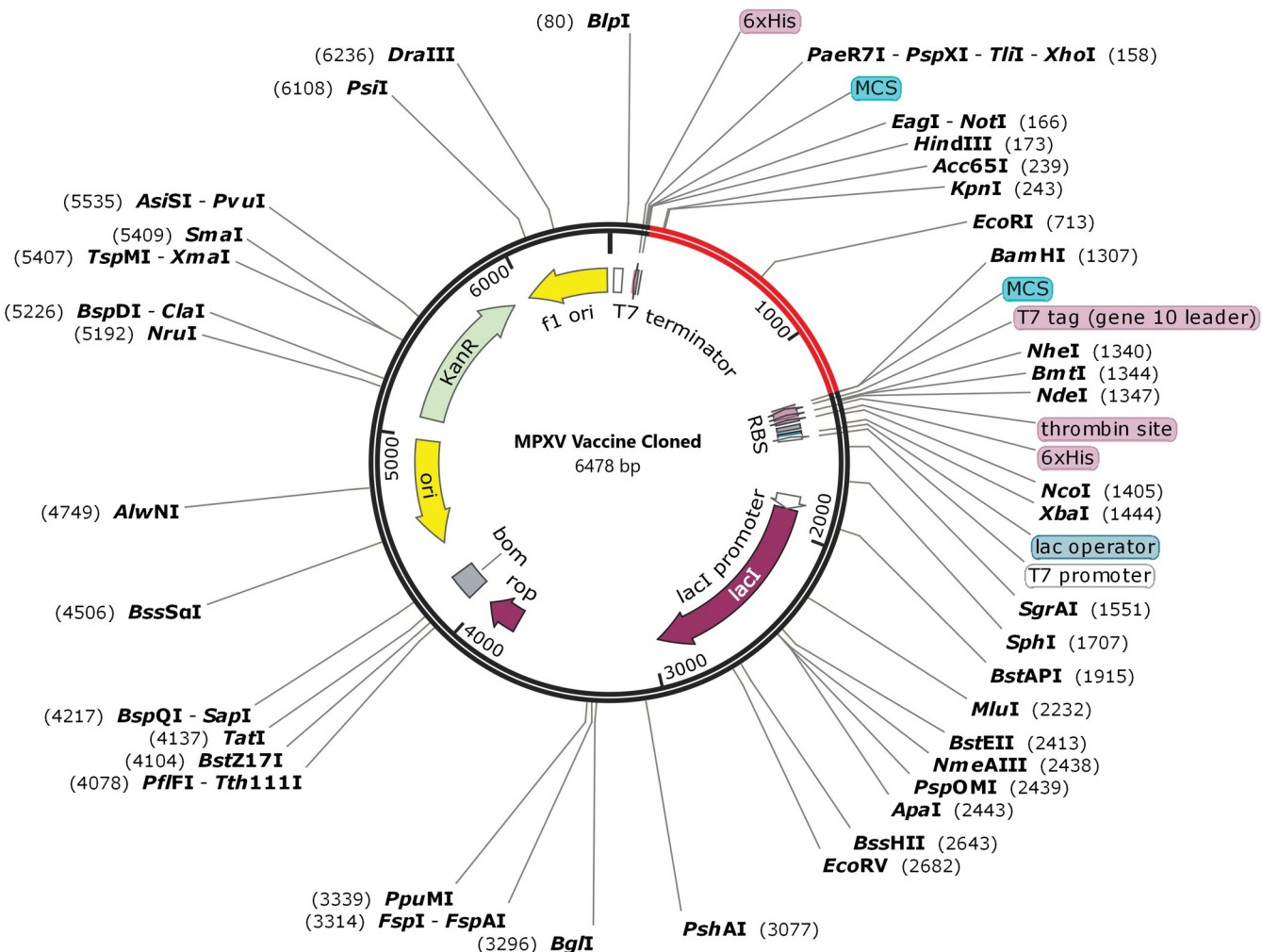

**Fig 11. *In silico* cloning.** The sequence of the multi-epitope vaccine (shown in red) surrounded between *HindIII* (173) and *BamHI* (1307) into the pET-28a (+) expression vector (shown in black).

protein L7/L12, b-defensin, LT-IIC, CTB, RS09) resulted in five different vaccine constructs [104]. In the study of Jin, *et al.* (2023) a vaccine construct was designed that included B- and T-cell epitopes from B13R, Abundant component of virosome, Bifunctional zinc finger-like protein/E, Ankyrin, DNA-directed RNA polymerase subunit, A18L, Bifunctional hemagglutinin/type-I membrane, IMV membrane protein, Interleukin-1 receptor antagonist-like protein, and CC-type chemokine binding protein. They also appended a beta-defensin-2 sequence as an adjuvant to the vaccine construct's N-terminal [105]. In comparison to published articles, our study is unique in that it combined the epitopes of the cell surface-binding protein and envelope protein A28 homolog for the first time with CTxB (adjuvant) to construct a multi-epitope vaccine against MPXV.

Because the cell surface-binding protein and envelope protein A28 homolog are important for virus infectivity, we considered them for epitope prediction [106]. Epitope mapping of the aforementioned proteins was performed, and epitopes interacting with at least three MHC class-I and MHC class-II alleles were chosen for further screening. We planned to design a universal multi-epitope vaccine against the MPXV. For this purpose, in addition to evaluating

epitopes for parameters such as antigenicity, toxicity, and allergenicity, as well as induction of IFN-γ and IL-4 (only for HTL epitopes), we also considered their presence in conserved regions from all virus strains available in NCBI. The multi-epitope vaccine was constructed by assembling the selected BCL, HTL, and CTL epitopes and appropriate linkers as well as the CTxB (as adjuvant). The CTxB is a non-toxic component of cholera toxin that has a high affinity for the monosialotetrahexosylganglioside (GM1) receptor, which is found in many cell types, including gut epithelial cells, antigen-presenting cells, macrophages, dendritic cells, and B cells [107]. As a result, CTxB has been widely employed as an adjuvant in vaccine design to boost immune responses [108–110]. In this study, like many other studies, the AAY linker was used to connect CTL epitopes [85, 111, 112], the GGGGS linker to connect HTL epitopes [113, 114], the KK linker to connect LBL epitopes [115–117], and the EAAAK linker to connect adjuvant [116, 118, 119]. The AAY linkers help to increase epitope presentation while decreasing the formation of junctional epitopes [120]. The GGGGS linker, which is made up of small or polar amino acids like Gly and Ser, provides good flexibility and solubility and is an excellent choice for fusion protein domains that require specific movements or interactions [121]. The KK linker is a target for cathepsin B, a key protease in antigen processing. This linker helps in the reduction of junctional immunogenicity by preventing antibody induction for the peptide sequence formed when the individual epitopes are linearly connected [120]. The EAAAK linker is a rigid alpha-helix-forming peptide linker that functions as a domain spacer in fusion proteins [122].

According to population coverage analysis, the vaccine construct covers 95.57% of the worldwide population, while the vaccine designed in study of Waqas *et al.* covers 93.62% of the worldwide population [102]. The high antigenicity scores predicted by the VaxiJen 2.0 server and the ANTIGENpro server showed that the proposed vaccine is effective enough to elicit strong immune responses in the body. Furthermore, the vaccine construct's non-allergic behavior was validated using the AllerTOP v. 2.0 server. The vaccine has good solubility, according to the Protein-Sol server's solubility score (0.455), which was higher than the threshold (0.45). The solubility score of the vaccine designed in the study of Shantier *et al.* (2022) was lower than the threshold, so in terms of solubility, the vaccine designed in the present study is preferable to this vaccine [101]. Evaluating the physicochemical properties of the vaccine structure provides us with relative knowledge of the vaccine's physicochemical properties, which facilitates *in vitro* and *in vivo* vaccine evaluations. The vaccine construct had a molecular weight of 41.97 kDa, which is ideal because proteins with a molecular weight of less than 110 kDa are easier and faster to express and purify than heavier proteins [95]. The proposed vaccine's theoretical pI was determined to be 9.48, implying its alkaline nature. Our vaccine candidate's half-life was estimated to be 30 hours in mammalian reticulocytes, more than 20 hours in yeast, and more than 10 hours in *E.coli*, indicating that the vaccine is exposed to the immune system for an extended period of time [123]. In terms of half-life, our vaccine outperformed the vaccine designed by Waqas *et al.*, which had a half-life of 7.2 hours in mammalian reticulocytes (*in vitro*) [102]. The half-life of our vaccine matched the half-life of the vaccine construct designed in the study by Ullah *et al.* [100] and Shantier *et al.* [101]. The vaccine instability index was determined to be 38.70, which is less than 40, indicating that the vaccine candidate is stable under standard conditions [123]. The vaccine's aliphatic index was computed to be 91.12, indicating that the vaccine is stable over a wide temperature range [124]. The aliphatic index value of this vaccine was higher than the aliphatic index of the vaccine designed in studies by Waqas *et al.* (74.63) [102], Zaib *et al.* (79.90) [103], and Shantier *et al.* (65.75) [101]. The GRAVY score of the multi-epitope vaccine was calculated to be -0.044; this parameter indicates hydrophilicity and is related to protein solubility, and its negative value indicates that the vaccine interacts better with water molecules [32, 125].

After modeling, the vaccine's 3D structure was refined to optimize its quality and bring it closer to experimental accuracy. The Ramachandran plot revealed that in the initial model, 83.188% of the residues were present in the highly preferred region, while after refinement, the number of residues in this region increased to 94.783%. The z-scores for the initial and refined models were calculated by the ProSA web server as -3.48 and -3.76, respectively. A higher negative value of this parameter indicates that the structure is of higher quality [22]. Humoral immunity along with cellular immunity, is critical in fighting pathogens in the body [126]. The B-cells which produce antibodies are responsible for the body's humoral immunity [127]. Discontinuous B-cell epitopes play a key role in the antigen–antibody response [128]. Our vaccine candidate has the potential to induce robust humoral immunity because it has a large number of discontinuous B-cell epitopes.

Various Toll-like receptors (TLRs) are involved in the initial interaction of host cells with invading viruses, regulate virus replication and host responses, and ultimately affect virus pathogenesis [129]. Hutchens et *al*. revealed that in vaccinia-infected mice, TLR4 recognizes a viral ligand rather than an endogenous ligand, and mice with a nonfunctional mutant version of TLR4 had higher viral replication, hypothermia, and mortality than control animals. The findings of this work demonstrated that TLR4 promotes a protective innate immune response against the vaccinia virus, which can help to develop new vaccines and therapeutic agents against poxviruses [130], therefore, we used TLR4 as a receptor in molecular docking analysis. The results of the molecular docking show that our vaccine has a strong affinity for TLR4 and can stimulate both innate and adaptive immunity. The RMSD plot revealed the vaccine-TLR4 complex's stability during the MD simulation The vaccine's binding to chain A of TLR4 has reduced chain A's flexibility in comparison to chain B, as shown by the RMSF plot. Furthermore, the high flexibility of the vaccine's amino acids 40–100 in comparison to its other amino acids can be attributed to the fact that this region has no interaction with the receptor protein and can move freely. The computation of the binding free energies confirmed that the TLR4-vaccine complex is more stable than each of its parts individually.

The CAI value and GC content of our vaccine were calculated by the JCat server to be 0.97 and 45.30%, respectively. CAI value between 0.8 and 1 [131], as well as GC content between 30% and 70% [128], are considered optimal for gene expression in the target organism. Although the findings of this study are encouraging, it is still essential to test the vaccine candidate both *in vitro* and *in vivo*.

## Conclusion

Following recent WHO reports of confirmed MPXV cases in non-endemic regions, global concern about the possibility of a new pandemic has been raised. There is still no licensed vaccine specifically for MPXV. Because computational methods allow researchers to develop an effective and safe vaccine in less time and at a lower cost, we used this strategy to develop a multi-epitope vaccine against MPXV in this study. We believe our vaccine candidate can be a universal vaccine capable of inducing cellular and humoral immunity because we incorporated the best T-cell and B-cell epitopes from the conserved regions of the target proteins into the vaccine construct. Furthermore, the proposed vaccine's high affinity for the innate immune receptor (TLR4) indicates that it could be able to stimulate both innate and adaptive immunity against pathogen infection. The findings of this study are encouraging, but they are based on computational methods that will never be able to replace laboratory validation; thus, we recommend that this designed vaccine construct be tested in animal models in the future.

## Supporting information

**S1 Data. Multiple sequence alignment of the cell surface-binding protein.**
(PDF)

**S2 Data. Multiple sequence alignment of the envelope protein A28 homolog.**
(PDF)

**S1 Table. The predicted CTL epitopes from the cell surface-binding protein.**
(DOCX)

**S2 Table. The predicted CTL epitopes from the envelope protein A28 homolog.**
(DOCX)

**S3 Table. The predicted HTL epitopes from the cell surface-binding protein.**
(DOCX)

**S4 Table. The predicted HTL epitopes from the envelope protein A28 homolog.**
(DOCX)

**S5 Table. The predicted LBL epitopes from the cell surface-binding protein.**
(DOCX)

**S6 Table. The predicted LBL epitopes from the envelope protein A28 homolog.**
(DOCX)

## Acknowledgments

The authors would like to acknowledge the Clinical Biochemistry Research Center, Basic Health Sciences Institute, Shahrekord University of Medical Sciences, Shahrekord, Iran.

## Author Contributions

**Conceptualization:** Mahdi Ghatreh Samani.

**Data curation:** Samira Sanami.

**Formal analysis:** Samira Sanami.

**Investigation:** Sajjad Ahmad, Muhammad Tahir ul Qamar, Hamidreza Pazoki-Toroudi.

**Methodology:** Sajjad Ahmad, Muhammad Tahir ul Qamar, Hamidreza Pazoki-Toroudi.

**Project administration:** Mahdi Ghatreh Samani.

**Software:** Shahin Nazarian, Shahram Tahmasebian.

**Supervision:** Mahdi Ghatreh Samani.

**Visualization:** Shahin Nazarian, Elham Raeisi.

**Writing – original draft:** Muhammad Tahir ul Qamar, Maryam Fazeli.

**Writing – review & editing:** Mahdi Ghatreh Samani.

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
