## [Decision Letter · Decision Letter 0]

18 Apr 2023

PONE-D-23-08627In silico design and immunoinformatics analysis of a universal multi-epitope vaccine against monkeypox virusPLOS ONE

Dear Dr. Ghatreh Samani,

Thank you for submitting your manuscript to PLOS ONE. After careful consideration, we feel that it has merit but does not fully meet PLOS ONE’s publication criteria as it currently stands. Therefore, we invite you to submit a revised version of the manuscript that addresses the points raised during the review process.

We look forward to receiving your revised manuscript.

Kind regards,

Sheikh Arslan Sehgal, PhD

Academic Editor

PLOS ONE

Journal Requirements:

https://assets.cureus.com/uploads/review_article/pdf/102530/20220810-30968-1qo1gw.pdf

https://www.researchgate.net/publication/345757265_Monkeypox_Virus_in_Nigeria_Infection_Biology_Epidemiology_and_Evolution

In your revision ensure you cite all your sources (including your own works), and quote or rephrase any duplicated text outside the methods section. Further consideration is dependent on these concerns being addressed.

   "This study was financially supported by the Research Deputy of Shahrekord University of Medical Sciences with grant number 6411."

4. Please note that funding information should not appear in any section or other areas of your manuscript. We will only publish funding information present in the Funding Statement section of the online submission form. Please remove any funding-related text from the manuscript.

   "Funding 

This study was financially supported by the Research Deputy of Shahrekord University of Medical Sciences 

with grant number 6411."

Reviewers' comments:

Reviewer's Responses to Questions

**Comments to the Author**

1. Is the manuscript technically sound, and do the data support the conclusions?

Reviewer #1: Yes

Reviewer #2: Partly

2. Has the statistical analysis been performed appropriately and rigorously? 

Reviewer #1: Yes

Reviewer #2: N/A

3. Have the authors made all data underlying the findings in their manuscript fully available?

Reviewer #1: Yes

Reviewer #2: Yes

4. Is the manuscript presented in an intelligible fashion and written in standard English?

Reviewer #1: Yes

Reviewer #2: No

5. Review Comments to the Author

Reviewer #1: This research study is very interesting "In silico design and immunoinformatics analysis of a universal multi-epitope vaccine against monkeypox virus", here author tries to determine the structure of epitope based vaccine designing against monkeypox virus . Following comments need to be incorporated in the MS

1. The introduction section is weak in terms of referring to the environmental aspects of Monkeypox virus, https://doi.org/10.1515/reveh-2022-0221 should be used to strengthen this section.

2. The authors chose the residues forming the disulfide bond based on a χ3 angle between -87° and +97° and an energy value less than 2.2 kcal/mol. Do you have a reference for these values? If you have a reference, be sure to give that cite.

3. In section 2.2. give the reason for evaluating epitopes for inducing the production of IFN-γ and IL-4.

4. Accession numbers of TLR4 and cholera toxin B subunit should be mentioned in the MS.

5. Authors should explain why they used cholera toxin B subunit as an adjuvant by citing the source.

Reviewer #2: The authors have used immunoinformatic approaches and designed a multi-epitope vaccine against monkeypox virus using the B and T-cell epitopes of the cell surface-binding protein and the envelope protein A28 homolog which has significance in the present research field. The authors have generated lots of data, but the manuscript is not well written. However, I would recommend major revision of this manuscript.

Comments

1. The abstract section is not well written. The abstract section needs more improvement with mentioning the specific finding (such as how may T-cell and B-cell epitopes etc.) were used for the vaccine construct instead of providing the overall findings.

2. The main limitation of this study is the data size. The authors have used only 10 sequences for the cell surface-binding protein and only 5 sequences for the envelope protein A28 homolog. The authors could use more sequences because the authors have depended on the conserved regions of these proteins. There are 100 of sequences are available in the protein sequence database such as Uniprot or in NCBI protein database. It is not clear why authors used only limited numbers of sequences among the large number of sequences?

3. Since this monkey pox virus is spreading in multiple countries, the analysis and showing the population coverage for the identified T-cell epitopes was important which is missing in this important study.

4. The authors could use 2 different servers for the allergenicity prediction for the cross validation of the allergenicity of the predicted epitopes.

5. The authors have mentioned about the in silico cloning in the materials and methods section, but they have not shown the Multiple cloning site in Figure 11 which needs correction and reanalysis of the in silico cloning for this vaccine construct.

6. The authors have predicted the three-dimensional structure of the final vaccine construct, so why did they predict the secondary structural features of the vaccine construct. Please provide a strong rational of using these or I will suggest removing this section from the manuscript.

7. The authors predicted the discontinuous B-cell epitopes from the Vaccine construct, but they have not discussed the significance of these findings.

8. The authors have performed the MDS for the TLR-4 and Vaccine construct separately (Figure 10), but it was important to perform the MDS for the TLR4-vaccine complex for the checking the stability of the complex. The authors could perform a 100ns MDS instead of the 40ns to see the proper stability of the vaccine, TLR4 or the TLR4-vaccine complex.

9. I would say the discussion section needs more improvement. In last few months, there are several research articles have been published for the multi-epitope vaccine design using immunoinformatic approaches in different journals. Could you please explain why this study is unique in comparison with the published articles.

10. Figure legends need to be more descriptive.

6. PLOS authors have the option to publish the peer review history of their article (what does this mean?). If published, this will include your full peer review and any attached files.

Reviewer #1: No

Reviewer #2: **Yes: **Utpal Kumar Adhikari

---

## [Author Response · Author response to Decision Letter 0]

9 May 2023

The authors of this manuscript thank the editor and reviewers for making suggestions and recommendations that helped us to improve the quality of the manuscript. We have made the changes suggested by the reviewers. 

Journal Requirements:

Ans: Thanks for the comments. All items were checked and corrected based on the PLOS ONE style templates.

https://assets.cureus.com/uploads/review_article/pdf/102530/20220810-30968-1qo1gw.pdf

https://www.researchgate.net/publication/345757265_Monkeypox_Virus_in_Nigeria_Infection_Biology_Epidemiology_and_Evolution

In your revision ensure you cite all your sources (including your own works), and quote or rephrase any duplicated text outside the methods section. Further consideration is dependent on these concerns being addressed.

Ans: Thanks a lot for your attention. The previous publications were addressed (reference 5 in line # 49 and reference 14 in line #73).

 "This study was financially supported by the Research Deputy of Shahrekord University of Medical Sciences with grant number 6411."

Ans: Dear editor, the above item was done accordingly.

4. Please note that funding information should not appear in any section or other areas of your manuscript. We will only publish funding information present in the Funding Statement section of the online submission form. Please remove any funding-related text from the manuscript.

 "Funding 

This study was financially supported by the Research Deputy of Shahrekord University of Medical Sciences with grant number 6411."

Ans: Dear editor, the above item was done accordingly.

 Ans: Dear editor, the above item was done accordingly.

Ans: It's done (line #268-270).

Ans: Thank you so much for your nice comment. It's done (line #962-978).

Reviewer #1: 

1. The introduction section is weak in terms of referring to the environmental aspects of Monkeypox virus, https://doi.org/10.1515/reveh-2022-0221 should be used to strengthen this section.

Ans: As per your valuable comment , this has been included in the manuscript (line #58-59 and reference 9).

2. The authors chose the residues forming the disulfide bond based on a χ3 angle between -87° and +97° and an energy value less than 2.2 kcal/mol. Do you have a reference for these values? If you have a reference, be sure to give that cite.

Ans: It was done accordingly (line #378-380 and reference 66).

3. In section 2.2. give the reason for evaluating epitopes for inducing the production of IFN-γ and IL-4.

Ans: Thanks for the comments. It was done (line #156-158 and reference 47).

4. Accession numbers of TLR4 and cholera toxin B subunit should be mentioned in the MS.

Ans: Thanks a lot for the attention. It was done (line #170 and line #231).

5. Authors should explain why they used cholera toxin B subunit as an adjuvant by citing the source.

Ans: As per your valuable comment , this has been included in the manuscript (line #531-534 and reference 108-111).

Reviewer #2: 

1. The abstract section is not well written. The abstract section needs more improvement with mentioning the specific finding (such as how may T-cell and B-cell epitopes etc.) were used for the vaccine construct instead of providing the overall findings.

Ans: Thank you so much for your nice comment. The abstract of the paper was thoroughly revised accordingly as the reviewer comment.

2. The main limitation of this study is the data size. The authors have used only 10 sequences for the cell surface-binding protein and only 5 sequences for the envelope protein A28 homolog. The authors could use more sequences because the authors have depended on the conserved regions of these proteins. There are 100 of sequences are available in the protein sequence database such as Uniprot or in NCBI protein database. It is not clear why authors used only limited numbers of sequences among the large number of sequences?

Ans: Thanks a lot for the attention. When we wrote this article, only these sequences were submitted and we only used them, according to your opinion, we also took the new sequences and analyzed them, and revised this part (line #120-124, line # 274-275).

3. Since this monkey pox virus is spreading in multiple countries, the analysis and showing the population coverage for the identified T-cell epitopes was important which is missing in this important study.

Ans: Thank you so much for your constructive comment. this has been included in the manuscript (line #173-178, line #310-318, line #545-546).

4. The authors could use 2 different servers for the allergenicity prediction for the cross validation of the allergenicity of the predicted epitopes.

Ans: Thank you for your valuable comment. If we use another server, we will lose some epitopes, and since the vaccine construct is designed, we cannot do this in this study, but we will definitely consider it in the next works. However, in many published articles, such as the articles below, one server has been used to determine the allergenicity of epitopes.

DOI: 10.3390/vaccines10050691

DOI: 10.1038/s41598-022-12651-1

DOI: 10.3390/ijerph19095568

DOI: 10.3390/vaccines10081300

DOI: 10.1038/s41598-022-14427-z

DOI: 10.1038/s41598-023-30445-x

DOI: 10.1016/j.virusres.2020.198082

5. The authors have mentioned about the in silico cloning in the materials and methods section, but they have not shown the Multiple cloning site in Figure 11 which needs correction and reanalysis of the in silico cloning for this vaccine construct.

Ans: This part and Figure 11 were revised based on the reviewer's comment, and the Multiple cloning site was marked in blue color in Figure 11 (line #257-266, line # 477-480).

6. The authors have predicted the three-dimensional structure of the final vaccine construct, so why did they predict the secondary structural features of the vaccine construct. Please provide a strong rational of using these or I will suggest removing this section from the manuscript.

Ans: Based on the reviewer's suggestion, this section was removed from the manuscript.

7. The authors predicted the discontinuous B-cell epitopes from the Vaccine construct, but they have not discussed the significance of these findings.

Ans: As per your valuable comment the above item were provided accordingly in the manuscript (line #576-580).

8. The authors have performed the MDS for the TLR-4 and Vaccine construct separately (Figure 10), but it was important to perform the MDS for the TLR4-vaccine complex for the checking the stability of the complex. The authors could perform a 100ns MDS instead of the 40ns to see the proper stability of the vaccine, TLR4 or the TLR4-vaccine complex.

Ans: Thanks for the your valuable comments. We investigated the behavior of the vaccine and the receptor in the docked complex, but their graphs were drawn separately and in some studies, it was done in the same way (DOI: 10.3390/vaccines10122010, DOI: 10.1038/s41598-022-11851-z, DOI: 10.1016/j.ijbiomac.2020.07.117,DOI: 10.3389/fimmu.2022.865180, DOI: 10.1016/j.sjbs.2021.06.082), , but your comment is very interesting and constructive and will definitely be considered in future studies.

After 33 ns, the vaccine reached an almost steady state and TLR4 also fluctuates very little during the simulation period. Due to the lack of facilities, this part of the work was done by a bioinformatics company, and due to financial limitations, we could only pay the company for 40 ns. I am sure you are aware of the bad financial situation of my country (Iran) and I hope you understand me. In many studies, such as the following studies, the simulation time is considered to be less than 40 ns (even 10 ns).

DOI: 10.1038/s41598-022-14427-z

DOI: 10.1021/acsomega.9b00944

DOI: 10.1007/s12026-022-09346-0

DOI: 10.1016/j.micpath.2020.104236

DOI: 10.1016/j.imu.2020.100478

DOI: 10.1371/journal.pone.0244176

DOI: 10.3390/vaccines10091564

9. I would say the discussion section needs more improvement. In last few months, there are several research articles have been published for the multi-epitope vaccine design using immunoinformatic approaches in different journals. Could you please explain why this study is unique in comparison with the published articles.

Ans: Thank you so much for your constructive comment. This part was added to the discussion section(line #505-522, line # 545-547, line # 560-562, line # 566-569). 

10. Figure legends need to be more descriptive.

Ans: Based on your valuable recommendation, Figure legends were modified.

---

## [Editor Report · Decision Letter 1]

11 May 2023

In silico design and immunoinformatics analysis of a universal multi-epitope vaccine against monkeypox virus

PONE-D-23-08627R1

Dear Dr. Ghatreh Samani,

We’re pleased to inform you that your manuscript has been judged scientifically suitable for publication and will be formally accepted for publication once it meets all outstanding technical requirements.

Kind regards,

Sheikh Arslan Sehgal, PhD

Academic Editor

PLOS ONE
---

## [Editor Report · Acceptance letter]

15 May 2023

PONE-D-23-08627R1 

In silico design and immunoinformatics analysis of a universal multi-epitope vaccine against monkeypox virus 

Dear Dr. Ghatreh Samani:

I'm pleased to inform you that your manuscript has been deemed suitable for publication in PLOS ONE. Congratulations! Your manuscript is now with our production department. 

Kind regards, 

on behalf of

Dr Sheikh Arslan Sehgal 

Academic Editor

PLOS ONE